# Study on the Relationships between Doctor Characteristics and Online Consultation Volume in the Online Medical Community

**DOI:** 10.3390/healthcare10081551

**Published:** 2022-08-16

**Authors:** Zhu Chen, Qingli Song, An Wang, Dong Xie, Huiying Qi

**Affiliations:** 1School of Nursing, Peking University, Beijing 100191, China; 2School of Basic Medical Sciences, Peking University, Beijing 100191, China; 3Department of Health Informatics and Management, School of Health Humanities, Peking University, Beijing 100191, China

**Keywords:** online health community, doctor characteristic, online consultation volume, emotional analysis, regression analysis, correlation analysis

## Abstract

Background. As a new medical service model, the online health community can integrate various medical resources to the maximum extent and improve the accessibility and utilization rate of hospital high-quality medical resources. Objective. Research based on the characteristics of doctors can enable doctors to display themselves on the network platform better, provide better services for patients, and improve the quality of medical services for doctors. Method. By crawling the characteristic data of doctors in Good Doctor Online, using dynamic analysis, correlation analysis and regression analysis, this study explores the relationships between each characteristic data and online consultation volume. Results. The doctor’s title and city level representing the static characteristics of the doctor have a weak impact on the doctor’s online consultation volume, and the doctor’s dynamic characteristics such as the number of patient completions, the number of gifts received, and the number of published articles can have a positive impact on the doctor’s online consultation volume. However, the recommended heat will negatively affect the online consultation volume, and the comment text has no significant impact on the doctor’s online consultation volume. Conclusion. Therefore, doctors should actively publicize and show their professional level and constantly optimize their dynamic characteristics, increasing the number of online consultations and thus improving their influence.

## 1. Introduction

With the deep integration of internet technology and the medical and health industry, online medical care, as a typical representative of the “Internet + Medical” service mode, is gradually becoming a widely adopted medical treatment mode. The online health community is composed of doctors, patients, and users seeking health information. It breaks time and geographical restrictions, integrates various medical resources to the maximum extent and gathers patients and doctors together [1]. The online health community integrates the advantages of various resources, expands the ways for tertiary hospitals to provide medical services, and improves the accessibility and utilization rate of high-quality medical resources in hospitals. It has certain positive significance for promoting the continuity of medical services and plays an increasingly important role in modern society [2,3,4].

There are many studies on online health community users, mainly on the health information behavior of online health community users. For example, Tao et al. studied the knowledge sharing behaviors of users in online health communities based on social impact theory and believed that online health community managers need to establish a supportive community atmosphere to alleviate users’ privacy concerns and promote users’ sharing behaviors [5]; Yuxin et al. combined motivation theory and social capital theory to analyze the influencing factors of the knowledge-sharing behavior of professional medical users in the online health community, and the results showed that altruism under the role of trust, shared vision, reputation and self-efficacy of health knowledge has a positive effect on the knowledge sharing behavior of professional medical users [4]. Based on consumer trust theory and consumer perception theory, Lu Quan et al. established an influence model of patients’ medical choice behavior in online medical communities. They analyzed and verified the influence of offline reputation, online reputation, service quality, contribution value, popularity, and price of doctors on patients’ medical choice behavior [6]. Yuying et al. used the trust source credibility model and trust transmission theory to study the influence of doctors’ attributes and online word-of-mouth on patients’ behavior in choosing doctors in online health communities [7]. Zhigang et al. analyzed the network structure and physician behavior of the e-health community from a social capital perspective. They concluded that the more connected physicians are, the more easily information is transmitted. At the same time, doctors with higher professional titles have obvious location characteristics, familiarity and trust, and higher reciprocity levels. This is more likely to influence the behavior of other doctors [8]. Studies on user trust and user relationships in online health communities, such as Hanmei et al.’s; found that users are more likely to accept suggestions when developing cognitive or emotional trust [9]. Audrain-Pontevia et al. investigated whether the online health community can strengthen the doctor-patient relationship [10]. Naiji et al. studied the impact of word of mouth on physician service quality on patient choice [11]. Yang et al. empirically studied the influence of doctors’ online reputations on outpatient experience sharing. They concluded that patients tend to share doctors with higher medical quality and service attitudes. Hospital online reputation positively (negatively) moderates the correlation between medical quality (service attitude) and patients’ propensity to experience after treatment. In addition, disease severity negatively affects doctors’ online reputation [12]. Jiang et al. constructed a model of influencing factors of users’ willingness to use information services in online health communities to analyze these influencing factors [13].

The research on online medical community users is mostly from patients’ perspectives, inorder to study patients’ medical selection behavior and the doctor-patient relationship.

The research from the perspective of patients takes patients’ information characteristics (such as behavior and speech) as the research object to analyze patients’ medical choice behavior, knowledge sharing behavior, trust relationship, willingness to use information services and related influencing factors, with the ultimate goal of better serving patients. The purpose of this study is also to better serve patients but taking doctors as the research object; the static description characteristics of doctors, dynamic behavior characteristics, and patients’ evaluation of doctors are studied to provide a reference for doctors’ services. However, research from the perspective of doctors’ user characteristics is less common. Exploring how doctors’ characteristics affect the number of online consultations can enable doctors to display themselves on the network platform better and promote doctors to improve the quality of medical services; at the same time, it is also helpful for patients to choose the right doctor to the greatest extent and to positively affect patients’ medical choice behavior.

This study takes the doctor, an important role in the online health community, as the research object. It is of great significance for optimizing the description of doctors’ characteristics, enabling patients to better understand doctors, improving the efficiency of diagnosis and treatment, and improving the quality of doctor services on online medical platforms.

The research is mainly carried out from the following aspects. 

The static description characteristic of doctor users reflects doctors’ regional distribution and professional titles. The more developed the area where the hospital is located, the higher the level of economic development and the better the overall medical conditions. Different professional title levels of doctors represent doctors’ professional level and authority. Doctors with higher professional titles need to accumulate more clinical experience and a higher professional level after a long practice time. We analyze the relationships of doctors’ cities and personal professional titles on online consultation volume.

The dynamic behavior characteristics of doctor users reflect the characteristics of doctors’ participation in communication in the community, which mainly includes the social characteristics of total visits, number of articles, number of patients, post-diagnosis evaluation, thanks letters, and gifts. After doctors provide consulting services to patients, patients can give feedback on doctors’ consulting services, voting or writing comments or thank-you letters which take a long time. In addition, they can spend money on buying virtual gifts to express their gratitude to doctors. On the one hand, the feedback behavior of patients expresses their gratitude to doctors. On the other hand, it also recognizes doctors’ professional trust and service process. At the same time, for patients who later choose a doctor, such information sharing can provide more information about doctors, improve other patients’ understanding of doctors, and help patients choose a more suitable doctor. We analyze the relationships of total online consultations, published articles, finished patients, diagnosis evaluation, thanks letters, and gifts on online consultation volume.

As an online health community is a doctor-patient interaction platform, patient evaluation can reveal the characteristics of doctors from a deeper level. In this study, the analytical method of the patient evaluation text is based on the emotional dictionary to determine the emotional tendency of the text and explore the relationships between patient evaluation and online consultation volume.

## 2. Theoretical Basis and Hypothesis

### 2.1. Signaling Theory

Michael Spence, the inaugurator of signaling theory, provided a signal model in his classic *Job Market Signaling*, indicating that it is difficult for employers on the side of information disadvantage in a job market with asymmetric information to grasp the actual level of applicants on the side of information advantage. To make the employers fully apprised of their productivity and capability, applicants usually send their academic qualifications to the employer as a signal to prove their ability to make up for the information asymmetry between them and obtain the final choice of the employer [14].

There is information asymmetry between doctors and patients in online medical communities, where patients are on the side of information disadvantage and doctors are on the other. According to signaling theory, doctors, as a party of information superiority, actively send signals to potential patients in online medical communities by creating their static characteristics. This quality information can be observed for free and will compensate for the information asymmetry between patients and doctors to a great extent. In addition, the higher the hospital level and doctor’s professional title are, the easier it is for doctors to improve their popularity and obtain a high online consultation volume. Furthermore, in the case of the same real information quality, the higher the online consultation volume, the easier it is to gain market recognition and gather patients or potential users.

From the contrasting position, the same is true. The inflow of many patient resources further promotes the quality of the online medical community. It lowers the difficulty of improving doctors’ recommendation rates, yielding a virtuous circle of more growth in the platform.

According to information economics, there is information asymmetry in the market. Usually, the seller knows more than the buyer. Mapping the online medical community, when looking for a doctor patient users mainly look at the static characteristics related to the doctor’s diagnosis and treatment ability online, such as professional title, hospital, city, and whether they have a high level of education. This information is usually explicit, which means users can judge it simply by observation before the consultation. Dynamic characteristics, including total visits, number of articles published, number of patients, diagnosis evaluation, letters of acknowledgment, and gifts, fully reflect doctors’ online communication status. The dynamic characteristics of doctor users are more excessive and difficult to observe in the short term. From this point of view, patients are more similar to buyers in the commodity market and are in a state of information asymmetry.

Based on the analysis above, we extrapolate that, with the increase in patients’ understanding of doctors and doctors’ social characteristics, such as hospital and recommendation rates, the degree of information asymmetry is reduced. At this time, the guarantee required by patients will become weaker, and the signals needed by doctors for guarantee will be reduced. Therefore, we propose the following hypotheses: 

**H1.** 
*The professional title of the doctor has a significantly positive effect on online consultation.*


**H2.** *A doctor’s city has a significantly positive effect on online consultation*.

**H3.** 
*The recommendation rate of doctors has a significantly positive effect on online consultation.*


### 2.2. Motivation Theory

From the philosophical level, all human behaviors result from the interaction between the individual and the external environment, and the source power of the interaction comes from individual needs. When demand is not met, it will encourage human beings to explore the necessary conditions and ways to meet the demand, producing motivation. In other words, all human social activities have hidden purposeful motivation elements, which stimulate or guide the behavior of the subject to move forward in the direction of realizing their own needs [15].

The generation of doctors’ dynamic characteristics in the online medical community can also be regarded as the product of the motivation subject in realizing the goal driven by demand. Therefore, the next question comes naturally—What are the motivations and demands of the online medical community? American psychologist Maslow divided human needs into five levels in his writing *A Theory of Human Motivation*: physiological needs, security needs, social needs, respect needs, and self-realization needs, and then expanded to seven levels, which is a gradual progression from basic needs to upscale needs. According to Maslow’s motivation theory, we infer that the need for respect and self-realization mainly determines doctors’ motivation.

Doctors’ motivation can also be divided into intrinsic and extrinsic motivation [16]. On the one hand, internal motivation mainly originates from self-concept, namely the action with the goal of self-realization. In this paper, we believe that doctors are willing to approach the goal of self-realization or professional value through information sharing. On the other hand, there is external motivation. The external motivation of doctors should mainly be practical and based on self-concept. Practical motivation refers to the actions taken by doctors to obtain rewards, such as rewards and gifts. In contrast, motivations based on self-concept refer to reputation improvement and status raising.

What is equally noteworthy is that online comments, as an essential part of the online health community, are a key way of reflecting the popularity of doctors. On various network platforms, online comments are generally considered a useful supplement to product page information, which reduces the difficulty for consumers to evaluate whether the product can meet their expectations. Schlosser’s research shows that online comments significantly impact consumers’ choices [17]. Projecting into the online medical community, patients, as consumers, will also be affected by the choice of online comments. Moreover, the intensity of the impact of positive comments and negative comments on patients’ choices is different.

Based on the analysis above, we make the following hypothesis: 

**H4.** 
*The total number of final finish patients has a significantly positive effect on online consultation.*


**H5.** 
*The number of evaluations after diagnosis has a significantly positive effect on online consultation.*


**H6.** 
*The number of thanks letters has a significantly positive effect on online consultation.*


**H7.** 
*The number of gifts has a significantly positive effect on online consultation.*


**H8.** 
*Positive comments have a significantly positive effect on online consultation.*


**H9.** 
*Negative comments have a significantly negative effect on online consultation.*


### 2.3. Social Capital Theory

Social capital is different from other forms of capital because it is a collection of real or virtual resources that need to be gradually accumulated in the social environment and is agreed inform or institutionalized. Social capital consists of real or potential resources embedded in individuals or social networks. In the online medical community, users’ medical health resources represent part of the social capital. Doctor users possess these in the form of professional medical knowledge, carrying over into published papers. Doctors also achieve the purpose of knowledge sharing by publishing papers.

Based on social capital theory and motivation theory, we hypothesize the following: 

**H10.** 
*The number of articles published has a significantly positive effect on online consultation.*


## 3. Methods

### 3.1. Data Acquisition

Good Doctor Online is one of the leading Internet medical platforms in China. By July 2021, more than 240,000 doctors had registered on the platform. The doctors registered on the platform are all certified by their real names, so they have confirmed doctor qualifications. In the traditional case, when patients select doctors, there is no reference from other patients, so patients choose doctors through doctors’ profiles. They are more likely to choose a doctor if their professional field is fit for the specific disease. The online medical community provides a platform for patients to make better choices through other comments. In addition, patients could upload their cases, post thanks letters, and give presents. After platform verification, others can view the comments. Usually, more thanks letters and gifts represent a more positive effect. In this study, we used the “Good Doctor Online” data to select doctor users from three different types of diseases (diabetes, leukemia, and depression). These three diseases are among the top three on “Good Doctor Online” platform. These common examples of chronic diseases, blood diseases, and psychological problems have typical clinical significance and research representativeness.

We designed a multithreaded crawler tool based on Python language to crawl the static descriptive characteristics of doctor users (including doctor geographical distribution, personal title, and seniority information.), dynamic behavior characteristics (including total visits, number of articles, number of patients, postdiagnosis evaluation, thank-you notes, gifts) and of patient evaluation data, with a total of 7409 data records.

The patient comment data crawled all comment data corresponding to the doctor user by using the SnowNLP program of the Python class library. A single statement was analyzed using the sentiment module to obtain a sentiment value corresponding to each comment. The size of the value represents the probability that the review is positive.

### 3.2. Data Preprocessing

We preprocess the raw data crawled by Python, obtain the data that meet the requirements of descriptive statistics through data cleaning, and then obtain the data that meet the requirements of relevant analysis and regression analysis through data conversion.

First, we performed valid field data matching, and each doctor’s registration ID was used as a keyword to match his/her online platform-related data. We used data matching to eliminate incomplete data cases to ensure doctor users had complete information.

Second, field assignment conversion is performed on the doctor user with complete information to digitize the text. The field conversion rules are as follows: the doctor title is divided into 4 grades from low to high, and the city grade is divided into 5 grades from high to low. The doctor title was converted according to the rule of resident = 1, attending physician = 2, deputy chief physician = 3, and chief physician = 4. City-data conversion refers to the retrieval of the city where the hospital is located by the name of the hospital and the classification and extraction of the city where each doctor is located. According to the city classification standard issued by the State Council, the city is classified and converted with digital codes, i.e., first-line city = 1, second-line city = 2, third-line city = 3, fourth-line city = 4, and fifth-line city = 5.

A preliminary normality test was performed on the cleaned data. The results showed that all independent variables were not strictly normally distributed, but the mean, standard deviation, skewness, and quartile could better describe the distribution. Therefore, descriptive statistical analysis was performed on some data to determine the overall distribution of variables in the data set. Table 1 shows the descriptive statistics. According to the table: 

The data of online consultation, published articles, completed visits, diagnosis, and evaluation, thank-you letters, and thank-you gifts of doctor users are highly discrete, among which the online consultation is the most discrete;

The average professional title of doctors is at a high level, and doctors with the title of deputy chief physician or above account for most of the data, which indicates that the level of medical resources in the online medical community is generally high;

Almost all the data show a skewed distribution, and the status of diagnosis and treatment by doctors in this community is different;

The discrete degree of the doctor recommendation index provided by the online community platform is relatively small compared with other data.

For the patient evaluation text and the effective field data matching, we used the stuttering word segmentation package to segment each sentence, read the emotion dictionary, and search the emotion classification of each word to calculate the emotion score. Moreover, we crawled all the comment data corresponding to the doctor user. We analyzed a single statement using SnowNLP using the Python class library and the sentiment module to obtain a sentiment value corresponding to each comment, wherein the value represented the probability that the comment was favorable. In this study, we randomly selected 100 reviews among all reviews for the above three diseases according to the proportion of the total reviews for the three diseases. For these 100 comments, we compared the results of program recognition to the results of human recognition to produce the results in Appendix A. Passing the manual identification test, we identified two critical points. The comment was considered positive when the sentiment value was greater than or equal to 0.8, negative when the sentiment value was less than or equal to 0.6, and neutral when the sentiment value was within 0.6–0.8.

Since the number of doctor users with comment text is relatively small, after matching the comment data with the static and dynamic data, the doctor users without comment text are deleted, and 3154 pieces of data can be used for correlation analysis and regression analysis. A normality test was performed on the 3154 finally cleaned data. The results showed that all independent variables did not obey a normal distribution.

### 3.3. Spearman Correlation Analysis

To prevent the multicollinearity of independent variables from affecting the results of multiple regression equations, because all the data of independent variables did not obey a normal distribution, we used Spearman correlation analysis to conduct correlation analysis on the cleaned data.

According to the independent variable correlation analysis (Table 2), there was a strong correlation between the diagnosis evaluation and the thank letter, with a correlation coefficient of 0.959. The total number of comments and the number of positive comments also had a strong correlation, with a correlation coefficient of 0.980. The correlation coefficients of other independent variables were within the normal range.

### 3.4. Regression Analysis

On the basis of correlation analysis, regression analysis was carried out on the cleaned data. At the same time, because the magnitude difference of data between independent variables is large, the data distribution is rightward, some data were 0, and the dependent variable does not obey the normal distribution. We performed log transformation on these independent variables to reduce the impact of excessive magnitude differences. We also performed log conversion on the dependent variable: doctors’ online visit workload due to nonnormal distribution and obvious right deviation of data. Finally, we performed regression analysis on the cleaned data.

To eliminate the effects of multicollinearity problems between two data groups with high correlations, we established two controlled trials. We recorded the diagnosis evaluation and the total comments for group A, while for group B the number of thanks letters and positive comments after diagnosis was recorded. SPSS was used to normalize the data, and regression analysis was performed. The results are shown in Table 3 and Table 4.

## 4. Results

The experimental results of the two groups were consistent, indicating that the model’s conclusions were reliable. The R^2^ values of the models were all 0.780, indicating that the independent variable could explain more than 78.0% of the change in the dependent variable, and the goodness of fit of the model was good. From the F statistic result of the analysis of variance (sig = 0.000 < 0.005), it could be seen that the linear correlation of the regression model was significantly established at the significance level of 5%, so the regression model was explanatory of the hypothesis.

Because the effects of positive comments, negative comments, neutral comments, total comments, and city on the dependent variable were not significant and had no statistical significance, the multiple regression equation models were as follows: 

Online Consultation Volume A = 2.807 + 0.182 × Article + 0.306 × Patient Final Finish + 0.105 × Diagnosis evaluation + 0.520 × Gift + 0.088 × Professional Title − 0.234 × Recommend Rate

Online Consultation Volume B = 2.770 + 0.182 × Article + 0.312 × Patient Final Finish + 0.526 × Gift + 0.077 × Thanks Letter + 0.076 × Professional Title − 0.191 × Recommend Rate.

## 5. Discussion

The regression analysis showed that gifts and patients’ final finish were the two indicators with the largest positive regression coefficient in the regression equation, indicating that the doctor’s patient completion amount and the gift number received could most affect his online visit amount. Other variables such as published articles could also affect doctors’ online consultation, but the effect was insignificant. However, the titles of doctors had little effect on doctors’ online consultations. Furthermore, the comments and the doctor’s city did not affect the doctor’s online consultation. A doctor’s recommendation heat negatively affects a doctor’s online consultations.

From the perspective of signal theory, doctors’ professional titles had a positive effect on online consultation, so H1 was supported. The doctor’s city had no significant effect on the online consultation, so H2 was not tenable. The overall recommended rate of the doctor negatively affected the online consultation, so H3 was not established.

From the perspective of motivation theory, the doctor’s patient completion amount and diagnosis evaluation significantly affect online consultation, so H4 and H5 are true. Both the thanks letters and the gifts received by doctors positively affected the online consultation, and the number of gifts greatly affected the online consultation of doctors, so H6 and H7 were true. The patient’s comments, both positive and negative, had no significant impact on the doctor’s online consultation, so H8 and H9 were not supported. The total number of comment texts and the number of favorable comments (i.e., positive comments) of patients had a weak negative effect on the online consultation of doctors, indicating that positive comments could not significantly positively affect the online consultation, which might be because the excessive number of favorable comments of doctors led patients to believe that the comments contained “False data”, which reduced patients’ trust in doctors, resulting in decreased consultation.

From the perspective of social capital theory, the number of articles published by doctors can positively affect online consultation, so H10 is true.

## 6. Conclusions

The characteristics of doctor users in the online health community are directly related to their online consultation volume. Through the correlation analysis and regression analysis of doctor users and the comment text extraction of patient users, we explore the degree of importance of various factors affecting the number of doctors’ online consultations from the level of both static characteristics and dynamic characteristics. This study enlightens doctors in the online medical community on what characteristics they can focus on improving to improve their online consultation volume.

Here, we draw the following conclusion concerning the analysis of our experiment:

Doctors’ static characteristics have a weak influence on their online consultation volume. Through the analysis of the regression model, we found that the grade of the city has a weak impact on the number of doctors’ online consultations. The professional title has a positive effect on the number of doctors’ online consultations, but it is not as significant as the effect of gifts and patient final finish. Because the medical and health strength gap between regions is not obvious, and almost all regions have advanced medical resources, patients care more about which doctors with stronger professional strength can cure their disease. Therefore, patients do not pay much attention to which city the doctors in the online health community come from but pay more attention to the doctor’s personal professional title. In the eyes of patients, a doctor’s professional title can reflect his years of service and professional level. However, compared with the patient final finish and gifts given by other patients to doctors, the influence of doctors’ titles is slightly less. Therefore, the static characteristics of doctors in online health communities have less impact on online consultation than their dynamic characteristics. Only a doctor’s professional title has a positive effect on online consultation.

Most of the dynamic characteristics of doctors have a significant positive impact on the online consultation volume of doctors, but the recommendation rate negatively affects the online consultation volume. Regression analysis shows that the number of patients’ final finished, gifts, and published papers greatly positively impacts the number of doctors’ online consultations in the online medical community. One possible reason may be that patients tend to choose doctors with high professional levels when choosing doctors, and the number of consultations with such doctors will be higher. Gifts represent patients’ satisfaction with the doctors’ diagnosis and treatment effect and mirror the doctor’s professional level; the number of published articles embodies the doctor’s knowledge and specialty. Browsing the doctor dictionary and catching an ideal doctor with a high level of finished patients, articles, or gifts, patients form a recognition of his or her high professional level and then make a final decision. Nevertheless, the recommendation rate harms doctors’ online consultation, which is likely due to the ignorance of patients who have a misgiving about doctors climbing on the bandwagon.

The comment text had no significant effect on the doctors’ online consultation volume. The results from regression analysis indicate that positive comments have no impact on doctors’ online consultations. While most of the comment text is positive, the number of all comments also has no impact as above. Exploring the causes behind this, we found that perhaps patients would naturally hold a perspective that consistent positive comments are somewhat exaggerated when they drop down the comment section and see all of the same text such as “great”, or perhaps due to the law of diminishing marginal utility, if each doctor has much praise, the influence of praise on patients’ cognition becomes small, resulting in the weak influence on doctors’ online consultation volume. In terms of negative and neutral comments, generally speaking, their amount is much smaller than that of positive comments, leading to patients not being in the mood to find them, so they also have barely any influence on the online consultation volume.

On the one hand, in line with the principle of curing diseases and saving lives, doctors should give all patients comprehensive answers and put forward the most appropriate suggestions for the patient. Doctors should pay more attention to improving their professional levels, actively publicizing and showing this, such as writing more answers to common diseases which concern patients and producing more popular science publicity and education content. At the same time, doctors should try their best to meet the reasonable needs of patients, and timely and effective answers to the patient’s questions, without delay. In addition, doctors should listen patiently and explain modestly. Patients should be communicated with in the right way and with appropriate language. Such as gentle tone, cordial tone, etc. Because being energetic and keeping a nice attitude are beneficial to making an impression on patients and getting good feedback from them, doctors can continuously optimize and promote their dynamic characteristics, and eventually increase their online consultation volume.

On the other hand, online medical communities also need to make some adjustments. First, the qualification and ability of doctors registered in the community should be evaluated to overcome the first hurdle for patients from the perspective of the third party. At the same time, we should recommend suitable and high-quality doctors to serve the patients. Second, patients should be able to see the information on all doctors as much as possible to make the most suitable choice for themselves. They should not judge the professional level of a certain doctor because of his ‘high heat’.

## Figures and Tables

**Table 1 healthcare-10-01551-t001:** Descriptive Statistics.

Variables	Mean	Standard Deviation	Skewness	Quartile		
				25%	50%	75%
Professional Title	3.173	0.865	−0.694	3	3	4
Recommend Rate	3.258	0.438	1.497	3.0	3.2	3.4
Patient Final Finish	230.590	742.529	7.094	0	10	107
Diagnosis Evaluation	41.113	115.445	10.239	3	9	28
Thanks Letter	16.155	52.188	13.344	1	2	10
Gift	40.618	170.394	14.237	0	2	18
Article	15.111	84.390	17.796	0	0	8
Online Consultation	701.367	1925.621	6.682	8	76	491

**Table 2 healthcare-10-01551-t002:** Correlation Analysis.

	RR	OC	Arti	PFF	DE	TL	Gift	PT	TC	PC	NgC	NuC	City
RR	1.000												
OC	0.563 **	1.000											
Arti	0.299 **	0.619 **	1.000										
PFF	0.586 **	0.806 **	0.509 **	1.000									
DE	0.736 **	0.816 **	0.492 **	0.821 **	1.000								
TL	0.676 **	0.817 **	0.504 **	0.819 **	0.959 **	1.000							
Gift	0.603 **	0.884 **	0.577 **	0.798 **	0.857 **	0.862 **	1.000						
PT	0.368 **	0.244 **	0.137 **	0.144 **	0.326 **	0.271 **	0.241 **	1.000					
TC	0.719 **	0.631 **	0.368 **	0.747 **	0.718 **	0.695 **	0.627 **	0.072 **	1.000				
PC	0.712 **	0.622 **	0.363 **	0.734 **	0.710 **	0.691 **	0.621 **	0.072 **	0.980 **	1.000			
NgC	0.403 **	0.349 **	0.209 **	0.417 **	0.398 **	0.364 **	0.339 **	0.059 **	0.509 **	0.425 **	1.000		
NuC	0.381 **	0.333 **	0.157 **	0.425 **	0.360 **	0.338 **	0.312 **	−0.013	0.529 **	0.457 **	0.268 **	1.000	
City	−0.256 **	−0.104 **	−0.069 **	−0.012	−0.217 **	−0.192 **	−0.170 **	−0.083 **	−0.061 **	−0.066 **	−0.042 *	0.016	1.000

** Significantly at the 0.01 level (bilateral). * Significantly at the 0.05 level (bilateral). RR = Recommend Rate. OC = Online Consultation. Arti = Article. PFF = Patient Final Finish. DE = Diagnosis Evaluation. TL = Thanks Letter. PT = Professional Title. TC = Total Comments. PC = Positive Comment. NgC = Negative Comment. NuC = Neutral Comment.

**Table 3 healthcare-10-01551-t003:** Regression Coefficient (Group A).

Variables	Unstandardized Coefficients	Sig.	Model Effect
Professional Title	0.088	0.000	R2 = 0.780 F-Statistics Sig. = 0.000
City	0.003	0.905	
Recommendation Rate	−0.234	0.001	
Patient Final Finish *	0.306	0.000	
Diagnosis evaluation *	0.105	0.001	
Gift *	0.520	0.000	
Negative Comment *	−0.026	0.309	
Article *	0.182	0.000	
Neutral Comment *	0.024	0.353	
Total Comments *	0.005	0.866	

Sig. = Significance. * The data are logarithmically converted

**Table 4 healthcare-10-01551-t004:** Regression Coefficient (Group B).

Variables	Unstandardized Coefficients	Sig.	Model Effect
Professional Title	0.099	0.000	R2 = 0.780 F-Statistics Sig. = 0.000
City	−0.003	0.890	
Recommendation Rate	−0.191	0.007	
Patient Final Finish *	0.312	0.000	
Thanks Letter *	0.077	0.008	
Gift *	0.526	0.000	
Positive Comments *	−0.004	0.876	
Negative Comment *	−0.020	0.410	
Article *	0.182	0.000	
Neutral Comment *	0.028	0.262	

Sig. = Significance. * The data are logarithmically converted.

## Data Availability

Opening data. were analyzed in this study. This data be collected on website (www.haodf.com, accessed on 10 March 2022).

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
