# Peer review of "Study on the Relationships between Doctor Characteristics and Online Consultation Volume in the Online Medical Community"

_healthcare, 2022, doi:10.3390/healthcare10081551_

Round 1
Reviewer 1 Report
Thank you for giving me the opportunity to review this manuscript on understanding the associations between doctor characteristics and online consultation volume in the online medical community. I enjoyed reading the paper, as it tackles an important topic and the results can provide practical implications. Nevertheless, I have some critical issues with the manuscript that I elaborate on below and that I would like to see addressed in a potential revision.
1) The authors highlight that this study focuses on doctor’s behaviors from the doctor’s perspective. It may be the point that this research is most differentiated from previous research conducted from patients’ perspective. However, I do not agree with the authors’ suggestion. Studying the correlation between the doctors’ characteristics and the amount of online consultation is ultimately helpful for patients’ medical service choices. What is the main difference between doctors’ perspective research and patients’ perspective research on this topic? It is difficult to find the novelty and significance of this research in the Introduction.
2) There exist inappropriate expressions on statistical analyses. On page 7, the independent variable correlation analysis looks very awkward. In general, correlation analysis let us know the association or the absence of the relationship between two variables. However, in correlation analysis, we cannot examine the causal relationship how an independent variable affects a dependent variable. This study performs a correlation analysis to test correlations between main variables and conducts regression analyses to examine the hypotheses. The authors need to use statistical terms accurately and carefully.
3) The discussion section of this study looks like the results section. In general, hypothesis verification based on the analysis results is performed in the result part (e.g., H1 was supported). In addition, the discussion section should not only summarize the results but also produce noticeable (theoretical and practical) implications according to the results. The present discussion lacks implications.
4) I recommend the authors to change the present title (study on the correlation between doctor characteristics and online consultation volume in the online medical community). This study focuses on exploring the causal relationships between doctor characteristics and online consultation volume by conducting regression analyses. Thus, the authors do not need to use the correlation word in the title.
5) According to the analysis results, doctors’ professionals title was positively related to online consultation. However, on page 9, it had little effect on online consultation. The authors need to check the report of results, expressions, and typos.
Reviewer 2 Report
This manuscript by Chen et al. described a study focus on the correlation between different characteristic data and online doctor consultations volume. The database is selected from one of the most popular online system, and the results were analyzed based on different criterias.
Based on their descriptions, here are couple questions:
1. what's the standard for the doctor to be registered as a "doctor" in the online system? How does patient choose the doctor based on their knowledge if the patient comment could not affect doctors' consulation volumn. And why the gift or thanks letters could be positive affect the consulation without any doubt of "false data"?
2. the format of the citation for authors' names are not consistant in the introduction section, eg: "Zeng Yuying" vs "Zhigang Li".
3. the table 3 and 4 title have differnt size.
Round 2
Reviewer 1 Report
This manuscript has been fully revised according to the comments and suggestions that I provided. Thank you very much for this revision.
Reviewer 2 Report
I have no further questions based on authors' responses.